# Songs of the Bauls: Voices from the Margins as Transformative Infrastructures

**Uttaran Dutta [1,*] and Mohan Jyoti Dutta [2]**

[1]   The Hugh Downs School of Human Communication, Arizona State University, Tempe, AZ 85281, USA

[2]   School of Communication, Journalism and Marketing, Massey University, Palmerston North 4442, New Zealand; M.J.Dutta@massey.ac.nz

*   Correspondence: uttaran.dutta@asu.edu

**Abstract:** Bauls, the rural minstrels who sing songs of transformation, are a socio-economically and politico-religiously marginalized cultural population from rural Bengal (both from eastern and north-eastern, India and from Bangladesh). They identify themselves outside of any organized religion or established caste system in India, and therefore are constituted at the margins of contemporary global South. Voicing through their songs and narratives of emancipation, they interrogate and criticize material and symbolic inequalities and injustices such as discrimination and intolerance (including class and caste hierarchies, and other forms of disparities) perpetuated by hegemonic authorities and religious institutions. Embracing a critical communication lens, this paper pays attention to material and discursive marginalization of Bauls and Fakirs, foregrounding voice as an anchor to communicative interrogation of structural and cultural inequalities. Through voice, Bauls and Fakirs foreground reflexive spiritual and humane practices that raise societal consciousness and cultivate polymorphic possibilities.

**Keywords:** Baul; Fakir; Bengal; voice

## 1. Introduction

> Mad, mad, we are all mad.
> ...............................
> Some are mad after wealth,
> And others for glory,
> Some go mad with poverty.
> Anonymous Baul Song (Bhattacharya 1969, p. 42)

Bauls, the wandering minstrels, are a cultural population from socio-economically and politico-religiously marginalized sections of rural Bengal (both from eastern and north-eastern India and from Bangladesh) (Salomon 1995). The Bauls[1] represent a polymorphic spiritual tradition, which historically was influenced by three religious praxis—*Sahajiya* (*Bajrayana* Buddhism), *Bhakti* (Hindu *Vaishnavism*, oftentimes embodied in the *Tantric* tradition), and Sufi/Dervish mysticism (Islamic *Marifat*), and perform their spirituality through songs as they travel around (Lorea 2017; Mondal 2013). Disrupting the religious traditionalism of the big religions, the syncretic practices of Bauls envisage transformations voiced in spiritually based disruptions of hegemonic traditions. Traditionally, these performances

---

[1]   Hindu and Muslim practitioners are called Bauls and Fakirs respectively. However, scholars oftentimes use the term Baul to talk about people who follow Baul, Fakir and other similar spiritual traditions.

and discourses are taught and shared orally, primarily to convey their philosophy and teachings within and outside their communities (Zecchini 2014). Bauls identify themselves as outcasts of any organized religion and the established caste system in South Asia, constituting the margins of the global South (Akter et al. 2017). Through performing their songs, they imagine alternate possibilities of emancipation from dominant intolerances and inequalities, and seek to raise consciousness towards fulfilling a "dream of an alternative, less-exploitative, more egalitarian spiritual and social order" (Urban 2001, p. 1087). UNESCO, on 25 November 2005, included the Baul tradition in the global list of "masterpieces of the oral and intangible heritage of humanity" (Khalid and Chowdhury 2018).

Bauls, like other spiritual-based anti-hegemonic folk performers across the globe, negotiate situated structural absences and discursive violence. For instance, the *Gaines* (folk musicians) of Nepal sing songs to depict their negations with material poverty (Harrison 2013); similarly, Sufi artists from Turkey and Middle East countries describe their experiences of facing severe threats from conservative Islamic institutions (Otterbeck 2008) and dominant institutions (the state, market structures) through songs. However, because of their interrogation of hegemonic forms of oppression, Bauls are systematically targeted, stigmatized, and harassed. Moreover, light-hearted Baul performances and/or folk-music competitions in popularity-based shows on televisions and other mediated platforms pose challenges to conventional practices of Bauls and Fakirs (Krakauer 2016). The proliferation of fake Bauls (with no spiritual and/or philosophical training or initiation) and urban folk-performers (from sophisticated upper class) and their commercialization of Baul songs threaten the survival of rural Bauls at the margins (Knight 2010). The ongoing marginalization of rural Bauls and Baul traditions are constituted amid market and religious forces that on one hand, co-opt Baul performances as commoditized aesthetics, and on the other hand, erase Baul articulations through ceaseless stigmatization and attacks on rural livelihoods.

Previous academic studies primarily focus on esoteric aspects of Baul discourses (including nuances and secrecy of sexo-yogic practices, and metaphoric connotations of seemingly non-spiritual phenomena like market or worldly artefacts); and, sometimes on utilization of Baul tunes to design messages for creating public awareness, and/or for corporate social responsibility activities. However, engaged research on Bauls' everyday negotiations with situated material and discursive adversities and on their agentic-negotiations of is almost non-existent. For instance, in their collection of Baul and Fakir songs, Dr. Shaktinath Jha and Dr. Upendranath Bhattacharya presented about 2000 songs (in two-volumes) and 670 songs respectively, where the number of purely non-esoteric songs is less than 1%. Among this handful of non-esoteric songs, only a few songs (approximately 0.7%) directly addressed the structural and communicative disparities at the margins (which constitute the focus of this article). As the overall socio-economic situation of lower-caste people (almost all the Bauls belong to lower-castes) of remote rural Bengal is mostly impoverished (e.g., 49% indigenous population and 21% *dalit* population in West Bengal, India live below poverty line (World Bank 2017)), previous research assumed and/or cursorily mentioned that socio-cultural and political situations of Bauls are a matter of concern. In other words, even if the oppressions and marginalization of Bauls received some attention, there is a need for focused studies to systematically understand their struggle against contextual odds as well as legitimize their fight for equality and consciousness raising towards imagining a transformed humane society.

In the contemporary cultural landscape when many folk-music traditions are under threat for a variety of socio-economic and/or politico-religious reasons in the global South, foregrounding voices and agencies of the marginalized performers (and traditions) is important in examining the ways in which plurality of world-views, rationalities and imaginings anchor social change communication (Dutta 2011). Particularly salient is the transformative role of cultural performances as sites of consciousness raising. The hegemonic religious traditions that perpetuate boundaries and are normalised through capitalistic agendas and logics are disrupted by counter-hegemonic articulations grounded in voice. This paper seeks to (a) document the discourses of the Bauls whose voices and narratives are continually marginalized and de-legitimized; and (b) foreground the agency of the Bauls on how they (i) address conditions of marginalization; (ii) communicate their commitment and actions towards raising consciousness; and

(iii) promote humanistic values. In doing so, it puts forth the concept of voice as anchor to emancipatory imaginings in bringing about transformation.

## 2. Baul Performances: Socio-Historical Context

The meaning and origin of the word '*Baul*' is debated and can be traced back to a few possible sources. They are (a) a Sanskrit word '*vyakula*' (impatient eagerness for god); (b) another Sanskrit word '*vatula*'(the inner flow of energy affected by wind, or madman (derived from Hindi 'baur'); (c) '*awliya*', an Arabic term, which refers to "holy man" or "saint"; and (d) the sahajiya Buddhist word 'bajrakul' gradually became 'bajul' and then 'Baul' (Mondal 2015; Salomon 1995). Most of the Bauls reside in West Bengal (mainly in the districts of Birbhum, Burdwan, Bankura, and in some parts of Midnapore, and north Bengal districts), and in north-eastern India; whereas most Fakirs live in the Nadia and Murshidabad districts of West Bengal, and in Bangladesh. They are primarily minstrels who travel from village to village and earn a living by performing. Sometimes they meet and live with other Baul practitioners in an *akhra/ashram* (a place where they share their views, songs, philosophy and teachings). *Madhukari* (subsisting on alms from begging and refusing anything surplus) is an important traditional aspect of Baul tradition, although many contemporary Bauls have alternative professions as a source of income (Krakauer 2016).

Historically, the colonial oppression and politics (of exclusion) of *Brahminical*, upper-caste Hindu and Islamic society created massive socio-political and economic uncertainty in the lives of the people of lower strata of society, which eventually gave birth to many "deviant sects" like the Bauls (Urban 2001). In the early nineteenth century, many such non-conformist and rebellious sects like *Aul*, *Baul*, *Sain*, *Fakir*, *Nera-Neri*, *Sabebdhani*, *Kartabhaja*, and *Kishoribhaja* proliferated throughout Bengal (Urban 2001). These 'lower' class (and caste) heterodox, "deviant sects" essentially questioned and communicatively challenged the colonial and *Brahmanical* rule, as well as religious dominance thorough their esoteric, subvert and critical, yet (often) coded, discourses (Mahbub-ul-Alam et al. 2014).

Though Baul tradition has its roots in various religious praxis, they differ distinctly from (and often reject) the assumptions, teachings and discourses of mainstream religions (Openshaw 2002). Baul tradition does not recognize any worship of (images of) deities, mosques, temples, religious symbols or sacred places as a part of their ideology. Bauls not only deny the importance of external religious practices or rituals, they strongly criticize the 'caste system' (the songs of Lalon Fakir, as well as many past and contemporary Bauls, addressed the issue of caste), and other forms of social inequality and injustice (Sengupta 2015). Such skepticism and protests in lower societal strata, specifically by the Bauls, came under heavy criticism and attack by both the dominant Hindu and Islamic authorities. For instance—'*Baul Dhangsher Fatwa*' (a *fatwa* for destroying Bauls and their tradition) was published by Maulana Riyajuddin Ahmad of Rangpur, Bangladesh at the end of the twentieth century. Likewise, in Kolkata, India, satiric songs were performed and processions were organized to denigrate Bauls and their practices under the patronage of Hindu orthodox leaders (Jha 2014).

For interrogating and non-conforming from the margins, the Bauls and Fakirs and their voices are historically delegitimized (if nor erased) by the hegemonic forces, including orthodox upper-castes, and dogmatic religious institutions (Das 2014). As an alternate communicative means to respond to dominant aggressions and intolerances, Bauls, oftentimes espousing a poetic and indirect way, adopt aesthetic expressions as entry points to discursive spaces (Lorea 2018; Novetzke and Patton 2008). As a part of their struggle with structural absences (such as oppression, discrimination, exploitation, and poverty etc.), they seldom sing evocative and esoteric songs "in order to express and to seek some kind of deeper spiritual solution to these economic ills" (Urban 2001, p. 1109). It is important to note that Baul songs are mostly spiritual and philosophical in nature (e.g., they focus on several *tattwas* [*tattwa*—theory or epistemology of] including "*deha-tattwa*" [body], "*prema-tattwa*" [spiritual love], "*guru-tattwa*" [spiritual teacher], "*gujhyo-tattwa*" [esoteric], "*srishti-tattwa*" [creation], "*param-tattwa*" [supreme or almighty], as well as on "*manasiksha*" [self-realization]) (Sengupta 2015), but on a very few occasions (as previously mentioned—less than 1% of mediated and published discourses), they sing

and talk about socio-economic issues such as poverty, inequality and injustice; this paper is primarily invested in studying such discourses.

Those discourses (including songs) are mainly associated with (i) the material and discursive crises of their everyday lives; (ii) their ceaseless struggle for existence and identity; and (iii) their imaginings for creating humane and just world.

*Neoliberal Transformation of Baul Spaces*

Even if the Baul songs get some attention and popularity among the educated and affluent middle class in West Bengal and Bangladesh, the struggle for survival and equality is still a reality for the rural Bauls. Sengupta (2015) showed that in Nadia and Murshidabad (and elsewhere), Bauls and Fakirs are still experiencing oppressions, life-threats, and inhumane treatments from religious fundamentalists. Academics such as Dr. Shaktinath Jha played an important role by organizing Bauls and Fakirs in their struggle for recognition and existence. In addition, market forces greatly affected the lives and livelihoods of Bauls and Fakirs. Proliferation of mediated spaces and their competition- and popularity-centric agendas, as well as the changing tastes of contemporary audiences (e.g., their preference for rhythmic (i.e., dance-friendly), and brisk songs (Krakauer 2016), pose challenges to the traditional practices of the Bauls. Such emerging tendencies are antithetical as well as detrimental for Baul ideology, and their conventional spiritual praxis, which ultimately reduces Baul songs to a form of commodity and a means of profitmaking. As a consequence, aspiring musicians, without any training or initiation in Baul tradition, are singing Baul songs as an attractive source of income; by creating competitions, such professional fake Bauls are making the survival of the Bauls even more difficult (Hanssen 2018). Moreover, the Bauls and Fakirs oftentimes are forced to perform in trains and other public gatherings, despite the significant health risks of singing in air-polluted and noisy environments (Krakauer 2016).

Embracing a critical communicative lens, this article argues in favor of foregrounding the voices and cultural expressions of the marginalized folk-tradition to bring about social justice, equity and transformation. A critical cultural communicative lens essentially questions the hegemonic portrayal of underrepresented culture and/or cultural expression as inferior, "deviant" and inappropriate, and thereby communicatively engages at the lower socio-cultural strata to recover the voices of the marginalized (Dutta 2015; Shome and Hegde 2002). Such discourses (often the voices of dissent) question the dominant agendas and oppressive practices, and the hegemonic construction of the marginalized as devoid of agency (Reed 2005). Scholars have noted that the voices and agency from the margins, and even the indirect or passive defiance, are the 'stubborn bedrock' of nonconformities and skepticism, and also the key organizing force to imagining social transformation (Chaturvedi 2007; Scott 1990). Harter et al. (2009) noted that, in marginalized contexts, artistic performances can serve as useful tools to question dominant discourses and oppressions, and to create transformative possibilities. They further stated that, "for groups that are silenced and systematically denied discursive space because of political conflict, aesthetic performance serves as a means to reclaim voice, and offer an alternate narrative of hope and peace" (p. 38).

## 3. Methods

We conducted an extensive web search and literature review to search for Baul- and Fakir-related resources. For Baul and Fakir discourses, several media files (various formats of media file, including audio and audio-video formats) were searched through internet search engines, video hosting platforms (e.g., YouTube, Vimeo) and web-archives. Key words used in the search included Baul, Fakir, Baul Gaan, Fakiri Gaan, Folk songs of Bengal etc. After a comprehensive search of online resources, we located approximately 2000 Baul songs, documentaries, movies and interviews, and categorized them into two main categories—Devotional songs, and Songs with a material connotation. For this study, we only considered the discourses (including songs) which were about human suffering, contemporary issues, oppression and caste-based discriminations. After careful listening and/or watching, approximately

75 songs, interviews and documentaries were selected for content analysis for this paper. Baul songs, as noted by scholars (e.g., Kuckertz 1975), are performed differently in different parts of Bengal; in terms of musical sounds, some of the prominent styles are *Bhatiyali* (eastern Bengal), *Jhumur* (western Bengal), and *Bhawaiya* (northern Bengal).

Although it is undeniable that musical sounds[2] and/or performing styles, as well as their diversity, is an important aspect of academic research, this manuscript primarily focuses on the lyrics of songs, and on the meanings or messages communicated through Baul discourses in order to study the aforementioned critical-cultural issues.

The collected media discourses (i.e., songs, interviews and documentaries) were primarily in the Bengali language and its dialects, were translated and transcribed verbatim in English for data analysis by the first author, and checked by the second author. While transcribing, the first author oftentimes experienced difficulties in translating local proverbs and colloquial expressions. In order to ensure the authenticity of the translation, local people and scholars from Eastern and North-East India were consulted during the translation process, and in addition, the transcriptions of the interviews were also examined by another academic who is well-conversant in both the Bengali and English languages (Lincoln and Guba 1985).

An approach following the guidelines of grounded theory was employed for analyzing the data (Charmaz 2000). The discourses (e.g., lyrics of the songs and transcripts of conversations) were carefully analyzed and several categories created. A constant comparison technique was used to compare and contrast the themes and concepts that emerged from the discourse. This process helps to understand various contextual aspects (e.g., discursive and structural marginalization) as well as foreground the voices and agencies of Bauls and Fakirs that seek to raise consciousness and build a humane world. This process yielded several open codes. Then the open codes were analyzed, and after that, axial coding and selective coding process were conducted.

Open coding was particularly useful in identifying discrete concepts; these concepts were then labeled and sorted. This step leads to identifying and organizing discrete concepts, where the data were examined sentence by sentence, which was the basis of concept development. Axial coding was conducted after the open coding process. In this stage, various discrete concepts were grouped, and a relationship-building exercise was undertaken; this process helps to relate the conceptual categories with groups of similar phenomena. Then through the selective coding process, theoretical integration was accomplished. Finally, several overarching themes were identified in the discourses (contents or messages of about 10–20 songs and several interviews are represented in each theme): Structurally constituted material injustice, communicative marginalization and hegemonic cultures, and emancipation, reflexivity, and transformative communication.

## 4. Everyday Counter-Hegemonic Performances

The enunciations of the Bauls embodied in the songs, interviews, and documentaries depict voice as everyday communication that questions dominant oppressions and/or ignorance, and legitimizes their (Bauls') ceaseless negotiations from the margins. Through the voicing of the structural and communicative absences and injustices, through the interrogation of the hegemonic practices of erasure, and through consciousness raising and reflexive performances, the Bauls and Fakirs open up avenues for emancipatory imaginings and transformative possibilities.

### 4.1. Structurally Constituted Material Injustice

As a medium of expression, Baul songs foreground embodied narratives of the experiences of being at the margins. For example, owing to prolonged structural inequality, Baul communities, (being members of lower socio-economic strata, and often from the low and untouchable castes (referred to

---

2  This project does not examine musical sounds; however, musical sound can be studied in the future projects.

as *dalits*)), experience struggles in earning the bare minimum requirements of living. Such structural injustices force them to face abject poverty and social exclusion, which have been further exacerbated by the transformation of rural livelihoods amid neoliberal reforms in India. Through their songs, they describe the unbearable conditions of their existence. Sambhu Sadhu, a Baul practitioner from the state of Assam, India, while sitting in his own house, sang,

> My life weeps in broken fragments (Alas!)
> Emotionally dumb,
> For there is such great dearth. . . .
> No support I have . . . apart from my quilt . . .
> As gluttony draws to mouth upon eating rasgullas [Bengali sweet],
> Vegetable rice is a dream of yore, wheat is what goes down the mouth,
> Rice is finished, the price of wheat has risen,
> No way for the poor,
> For there is such a great dearth.

Thus, in their songs, the Bauls often talk about their negotiations with the lack of access to material resources, voicing their economic and socio-cultural struggles amidst the transforming landscape of India (note the narratives of inflation and price increases voiced here). The enunciation of the structural the marginalization of rural life is juxtaposed amid the voicing of the embodied struggles of the Baul to feed her/himself.

Their articulation of material scarcities and absences are not always forthright in nature, since sometimes they use indirect and layered communication. For instance, oftentimes in their voicing of material struggles, Bauls draw on spiritual discourses, mentioning the names of gods, to explain their situation poetically. In a song, Nakshtra Das Baul said,

> Some go to the market and stare,
> While some buy honey and sugar,
> Azure-throated God (Shiva) buys cheap vegetables and gourd,
> I cannot even buy a single thing. (Folkpick 2010)

Through these narratives they enunciate their day-to-day negotiations with the hegemonic market forces. They point to the marginalizing forces of the dominant market logics that have resulted in the misdistributions of wealth in society, and its scarcities at the margins. The juxtaposition of spirituality amid their voicing of everyday struggles resists dominant (such as neoliberal) intentions of greed and material accumulation.

Bauls opine that the most privileged people do not even have the time and willingness for listening to and/or addressing the situations of marginalized Bauls. Although their voices are largely unheard, their ceaseless struggle for survival, and their frustrations, often are revealed through their words and songs. A Baul singer from North Eastern India sang,

> What shall I say, who would listen?
> Will come forward any eager ear? . . .
> Spent my days in starvation
> Even as rice grows abundant on our earth.

Such struggles and negotiations are not only limited to the individual realm; socio-economic disparities, and inequity in the broader societal contexts are also voiced in Baul songs. One such song poetically narrated, "Fire is everywhere in the water, all the vegetables get boiled due to that, vultures are in the air, are coming to attack, *Hari-Narayan* (Lord Vishnu) sank inside, . . . everything is haphazard." Along with elaborating on the realities of the underserved in the contemporary market-dictated world, the Baul songs also show that economically poor people, and even the gods of the poor, could not overcome the scarcity of resources in their everyday lives. A Baul singer from Birbhum, West Bengal, sang,

> I found neither a friend nor even a lover,
> When I asked for happiness, misery is all I got
> We poor had hopes from the Almighty
> But, when we found him, he too was poor like us.

In some sense, the Bauls in their constant negotiations at the margins poetically depict potential impossibilities in overcoming adversities in their struggle for survival; they do so by showing that even for gods, such overcoming is unattainable. The depiction of the life of the poor also serves as an anchor for spiritually situating poverty.

Structural injustice not only materially affects impoverished populations, but also makes them socially alienated and isolated from mainstream society. Such de-legitimizations cause their voices and realities to be ignored and overlooked in the spaces of discursivity. In a Baul song, Kubir Gonsai depicted such realities,

> The hawks are flying in large numbers,
> Swooping down in a moment, they would come,
> On the branches would alight,
> They would feed on the flesh, picking out our eyes.
>
> Of this misery whom would I tell,
> Everyone would dine on my suffering well,
> Ruin me after and destroy beyond tell.
> The constant pain in my mind,
> To whom do I go, to whom confide,
> Neighbors and well-wishers are not in sight.
> Giving everything, the heart still pains, still yearning,
> They always hurt us, by cruel words they do burn us.

The song not only talks about oppressions, exploitations and helplessness at the margins, but also notes the mental stresses and pains experienced by the Bauls. Such an isolation and/or disenfranchisement both further marginalize the Bauls, and make their struggle for survival ever tougher.

As performers, they seek to secure respect and acceptance in their society. However, oftentimes, the experience of performance is daunting for many Baul singers. Subhadra Sharma, a woman Baul, shares her personal experience,

> In spiritual gatherings, we are made to sing the whole night, we never got notice from anybody. Even we also have hunger, but nobody paid any attention to that. Our performance would start at 7 or 8 p.m. in the evening and would go on until 4 a.m. in the morning. Nobody even thinks that we could be hungry. Our entire body would shiver from exhaustion. Tell me, what will happen to the person who is singing unfed? Ultimately they will become sick like me. A folk artist will eventually die from indigestion, gastritis. I think it is better to beg than to sing Baul songs. My sickness brought forth this epiphany,
>
> "As long is youth,
> So long is zeal,
> So long is desire and fire,
> Then there is quaking,
> Followed by shattering,
> Concluded in spinning." (Kaahon 2014)

Being members of the lower socio-economic strata, and having no sustainable income, most Bauls face exploitation and ignorance from the mainstream. Moreover, such a difficult journey oftentimes negatively influences their physical and psychological health.

For similar reasons, many of the Bauls stopped performing and opted for alternate income options. Scholars opined that the main reason for those who had left the arts, especially among aging practitioners, was that they could not provide sufficiently for the family from their low and/or uncertain income and resources (Jha 2010). Such marginality also made Bauls and their ways of life unattractive to their next-generations; for example, the son of late Kalachand Darbesh stated (the interview was conducted when he was alive),

> Sorrow will stay with us forever. Till now, I always find him (his father) in tears; I have never seen a moment of material-happiness. Today also, he has to toil hard for livelihood. Till today, my father goes by begging alms in the trains and attending fairs. I burst into tears seeing his struggles even in this age of life. It happened many a times that he was saved narrowly from death, when rescued from the streets.(Sahajiya Foundation 2013)

Like the late Kalachand Darbesh, many senior Bauls are struggling to meet the minimum needs of life in their old age. Situated resource scarcity, hunger, psychological stress, and that too with no social security, make their journey strenuous. This is more apparent in this contemporary era, where any respect and recognition of ascetic lifestyles is gradually eroding. According to Parvati Das Baul,

> I will be definitely happy if the Baul community is looked after well, especially the old masters. When they are sick and cannot perform, they have no value (in the societal and/or material sense). They have to just depend on whatever is coming (as alms); sometimes the situation is very, very pathetic. So many Bauls I have seen, they do not even have money to buy medicine. So, if these things are taken care of by the government, then it would be great.

In the contemporary era of increasing financial uncertainty, external aid and assistance are certainly helpful and important for Bauls. However, Bauls traditionally practiced *Madhukari* (i.e., everyday ritualistic begging for alms by walking from one village to another) for survival and sustenance. Changing economic, social, and cultural scenarios in this neoliberal era mark decreasing patronage for the Bauls; consequently, both the nature and practice of *Madhukari* are also changing. Unable to earn basic minimum subsistence because of the rapid liberalization of rural spaces, the Bauls are now performing in public places and/or gatherings, including in trains (aforementioned song of Nakshtra Das Baul is an example of such a performance). One of the senior Bauls, Paban Das Baul was talking about the contemporary practice of *Madhukari*, "The conditions are different now. These days, Bauls sing in trains. Do you know why? Trains do not make for a good setting. You can hardly hear the songs. They sing for only one reason—to survive." In contrast to visiting rural households on a regular basis with an aim to awaken them spiritually, many Bauls now have to entertain people in crowded spaces to earn food and essential commodities for their families. Their discourses highlighted the socioeconomic disenfranchisement experienced by Bauls; simultaneously, their utterances invert the logics of commoditization, greed, and profiteering. The articulation of the struggles with material resources are juxtaposed in the backdrop of the deep interrogation of economic attachments and market principles, drawing on a spiritual narrative.

Again, the questions and concerns about the survival of the Baul tradition, and the endangered existence of the Bauls, are also represented in Baul songs. For example, in anticipation of such danger, the famous Baul, Purna Chandra Das, once sang,

> Who will sing Baul songs?
> After experiencing so much pain or sorrow the Bauls have died . . .
> The bamboo-flute becomes speechless; so, Baul cries,
> With this sorrow (I would like to say),
> People have forgotten the songs of Bauls.

The song poetically captures the struggle of Bauls for survival, as well as an anxious anticipation of extinction by the Bauls themselves. However, by negotiating with all the aforementioned challenges,

disparities and adversities, the Bauls are relentlessly working towards overcoming material and discursive marginalization.

### 4.2. Communicative Marginalization Amid Hegemonic Cultures

Along with structural barriers and economic oppressions, Bauls experience various forms of symbolic marginalization–culturally, politically, and religiously. These various forms of marginalization have prolonged historical contexts and connotations, often tied to their lower caste or untouchable status.

Even before the era of Lalon, i.e., in the eighteenth century, this community was constantly facing opposition and oppressions from the dominant politico-cultural, as well as religious forces. These oppressions reflected the interplays of caste practices and practices of communicative marginalization. To date, the religious authorities portray performances and practices of Bauls and Fakirs as indecent and irreverent, and oftentimes label their songs as sinful (*haram*). About the religious-appropriateness of singing, one conversation was depicted in a biographical movie on Lalon Fakir—Moner Manush (released in 2010), where Lalon was answering one such question posed by his disciple:

> Disciple: Sai (Master), I have a question for you. Because we sing songs, many Muslim religious leaders are angry with us. They say bad things about our practices. They even beat us. They say that singing in Islam is a sin (*gunah*).

> Lalon: If the religious leaders say that singing is a sin, do not argue with them. By arguing, you will not reach anywhere. Holding your ears (i.e., admitting your mistake publicly), tell them, 'Master, I have made a mistake, please forgive me, I will not sing songs again in front of you'. Then keep singing inside. Those who are spiritually thirsty, keep serving music-nectar to them. (BongoBD 2015)

Communicatively speaking, Lalon in the aforementioned conversation advocated for strategic and veiled communication. His teachings argued in favor of apparently non-confrontational communication, which embodied an essence of ceaseless resistance (and a disapproval of mainstream religious praxis). We can describe such conversation as an example of 'stubborn bedrock' as delineated by Scott (1990).

Baul philosophy does not subscribe to a narrow definition of religion; it essentially questions the very intentions and dominant notions of religious organizing. Again, the Bauls put emphasis on true spiritual realization embodied in everyday experiences that can be achieved by anyone [and particularly as modes of spiritual possibility for excluded populations]. Therefore, they criticize hegemonic interpretations of sacred texts for misleading and controlling common followers. In an interactive Baul song (where two singers [one male and one female] are seen conversing with each other: One was favoring dominant discourses, and other one was talking about Baul philosophy), the female singer who was representing dominant discourse commented, "You (the Bauls) people don't obey the rules of the holy *Quran* and *Hadith*, How can you claim that you are Muslim? . . . You people always skip *Namaz*, In *Ramadan*, you do not fast . . . you people do not even participate in *Hajj* and other Islamic activities. If you want to go to heaven, fear the Allah". In response, the male Baul singer enunciated counter-arguments by saying, "By reading two pages (of religious text) it is not possible for one to grasp the spirituality of Bauls. They (religious leaders) do not even understand (the key-texts), and how can they teach people? By simply misinterpreting and misleading others, the Mullahs (religious leaders) take advantage. We are neither greedy for heaven, nor believe in the existence of hell. So (my friend) just introspect and nurture love inside you" (Tareque Masud Memorial Trust 2016). In the conversation, the first singer put emphasis on the external and/or ritualistic aspects of religious practice, such as reading sacred texts and participating in payer or fasting. The second singer, who represents the Bauls, challenged such ritualistic notions, and argued in favor of experiential realizations and spiritual attainments. Note here the juxtaposition of the Baul philosophy of syncretism in the backdrop of hegemonic religious orthodoxy.

The Bauls not only reject the dominant religious doctrines, but have also raised their voices in the context of social injustices, such as the existing caste system in South Asia. Thus, fundamentally,

they articulate their position against the hegemonic meaning-construction and knowledge production processes, working through their marginalized positions to disrupt hegemonic religio-social structures. In a documentary—Songs of the Bards of Bengal: The Bauls and Fakirs—Dr. Sudhir Chakraborty, argued,

> They (Bauls) offer their worship within themselves. Their rozah (ritualistic practice of abstaining from food and drink) is not physical fasting, but is a metaphoric expression of self-restraint and control of senses.

> Thus, when they chant the name of Hari or Allah, they actually mean something different; they refer to an internalized awareness.

They essentially embrace a spiritual ideology that dismantles the silos of religious orthodoxy, and envision humanity, consciousness and reflexivity at the center of spiritual accomplishment.

Bauls fundamentally reject mainstream religion and rituals, and perform alternate spiritual practices. In their spiritual path they place no importance on visiting places like temples, mosques, and idols, and reject the guidance of Brahmins or priests. In one song, Radhabalav, a Baul poet, communicated a similar message,

> Beguiled by the opportunists,
> You have lost your direction,
> Alas, which form of sadhana will then,
> Will fetch you the divine wealth?

> If Allah could be found in the streets of Mecca,
> Or Lord Shiva's residence was only in Varanasi,
> If Vrindavan was indeed the garden of Lord Krishna,
> Then none would have returned to their homes.

> If God could be had, With offerings of food—this and that,
> Or by proffering sinni[3], on a large scale,
> Then food would have been all it took,
> To have the King shook,
> To buy His amazing grace and escape his rebuke.

The song fundamentally questions and challenges the mainstream religious practices and rituals as well as argues for internal awakening and realizations. Chakraborty (2001) noted that scriptures, hymns, sacrifice, penance, pilgrimage or Holy Water is not important for Bauls and Fakirs; they search for their god among human beings; and their gurus are their initiator and pathfinder in the journey. In the path of alternate spiritual learnings, Gurus or *Murshid* (spiritual masters) hold a very important position for the Bauls and Fakirs. It is the gurus, who intimately and ultimately guide them. As Monimohan Das said in one of his songs—

> Oh Guru! You are Brahma, you are Vishnu, you are the revered god Shiva,
> Have mercy on me Guru, kindest thou be than all noble hearts, bless my soul and guide me over,
> You are help to those who are helpless,
> You are the Father and the keeper of us all,
> Stability to them you provide who have lost direction.
> Guru, I am all bereft,
> No path for me awaits, nothing for me in this world except for some shelter at your feet.

---

[3]    Sinni: An Islamic offering, made with different food ingredients, proffered to appease the Divine.

Gurus or their spiritual masters not only guide them in their spiritual journey, gurus also show them a path in their everyday negotiations with material and communicative adversities. Being members of the lower socio-economic strata, many Bauls do not get the opportunity to pursue and/or continue formal-education. Moreover, in spiritual and moral education contexts, they express their full faith (if not surrender) to their Guru, who is their mast in their spiritual journey. Chakraborty (2001) opines that they can even lay down their lives for gurus. Bauls come from different castes, classes, or religious groups. In their spiritual path, Bauls go above and beyond their boundaries of religion and silos of rituals; they express their faith in human consciousness and realization. One such Baul song notes,

> Be he Hindu or Muslim, be he Shakta, Buddhist or Christian.
> In the world of free love, everyone is equal.
> Those awakened by the knowledge of tattwa (epistemologies and theories),
> For them, darkness fades forever.

Through their songs, Bauls advocate for religious harmony, and celebrate the essence of core values and teachings of spirituality. To Bauls the almighty is not a mono-religious construct; they envision inclusive and equal entry of all people, irrespective of their individual identity or affiliation. A Baul song, narrates their vision—

> He (god) is the epitome of the Vedas and the Upanishads,
> He holds the Koran and reads the Namaz.
> The Christians think Him to be Christ,
> Creating inspirations of spiritual love and passions.

The song seeks to search the almighty in pedagogies and ideologies embedded in various religious traditions. Such utterances of inclusivity are important in this contemporary era of religious intolerance. Jha (2010) showed that Bauls come from the margins of Muslim, Hindu as well as Christian societies. As they are from underprivileged classes (including *dalits* and indigenous background), mainstream society portrays their practices as gounodhormo (marginalized religion), upodhormo (sub-religious), or opodhormo (Mal-religious). In a way, Bauls' long oppressive, dis-respected, unacknowledged existence make them (Bauls) anti-brahminical and counter-hegemonic. Further, because of their (Bauls') non-conformist attitudes and rejection of mainstream religious doctrines, they (Bauls) are also denied recognition in the mainstream. More specifically, mainstream religious institutions consistently posed restrictions, bans and/or even announced '*Fatwa*s' against such alternate practices to further marginalize the Bauls and Fakirs (Jha 2014). In one such case in the Murshidabad district of West Bengal, Jha (2014) documented a song that narrated the incident vividly. In February 1973, local political and religious leaders intentionally humiliated and ostracized Bauls publicly. Khodabaksh Fakir, writer of the song, narrated,

> . . . They feigned a meeting and invited us,
> "Come, my brothers to hear this meeting," he (a local leader—Golam Ali Murtuja) amiably said.
> A sacrifice of the Fakirs was to be made, it was told,
> Hearing this, people came, all young, as well as the old.
> Went there then the party all, curious to see what was going on,
> Some Fakirs were captured and made to swear on their religion,
> That scoundrels were they who indulged in marijuana's grave intoxication.
>
> . . . Fear of beating and lynching too,
> Made the Fakirs swear over and anew,
> Alas, but they do die at heart.
>
> . . . Near the banks of Sura (a river) there lived Panchu Shah (a Fakir),
> Now because of such trouble, chances of living he had none,
> Sitting in his home, he prays to the guru, deeming that he is the only hope.

> Seeing and hearing it all, some Bauls took heed, and to Beldanga (location of the incident) rushed,
> Orders were given, curfew in effect, which in turn Section 144 ushered.
> Some immediately fled, but three Fakirs were arrested.
>
> Manik Fakir and Ibrahim (a Fakir) have no more days to them to live.
> Mantu Khan (a local leader), Pretending to be a doer of good, cut off their (Bauls') buns off their heads,
> Chaos and outrage spread . . .

Though not common in the Baul literary tradition, this song is not only a testimony of Bauls' material and discursive marginalization, but it also demonstrates that through singing of the song, Bauls still remember and legitimize the incident among a wider audience.

In a similar tone, Baul Shah Abdul Karim of Bangladesh shared in one of his interviews, "People (*Sharia* followers) threatened that as I sing songs, they said that they will not bury my body in my religious custom." He then commented, "I committed many mistakes. We all make mistakes in every step of the way. But, my singing was not a mistake. I had never thought it was a mistake. It is my songs that brought me closer to the people's heart. It is my songs that created a consciousness within my being." Such expression of commitment and non-conformity is also reflected in the words of the younger Bauls; they refuse to submit to the mainstream doctrines and/or impositions; Bedana Fakir, a woman singer from Nadia, West Bengal answered,

> Interviewer: Do the Mullahs or those who abide by the Sharia law oppose you?
> Bedana Fakir: Yes, they do. In the beginning, they opposed us becoming Fakirs. But my guru refused to obey their diktat.

The mainstream authorities often confront Bauls' alternate spiritual journey. Moreover, compared to their male counterparts, female Bauls experience more opposition from patriarchal Bengali society; they are not supposed to go outside to visit homes or families. A woman singing songs in public is seen as blasphemous in many parts of rural Bengal; Mallika Akar, a woman Baul from Murshidabad said, "in our Muslim society, there is an impasse about women singing songs. [The men] don't let us step out of the house; even listening to songs is prohibited". Sufia Begam, a performer, in another interview shared what her family members told her, "You are a girl. Why do you need to go out in public? Why do you have to cast aside all your shame? You should remain at home. You will get married and go to your husband's house and look after the family. Why do you need to go out in public and sing?" In other words, Bauls continuously have to face many hurdles in their lives, if not the threat of extinction. To counter the alternate spiritual practices and epistemologies, the dominant institutions defame and misinterpret the discourses and ideologies of the Baul community. While negotiating with material and discursive marginalization, the Bauls and Fakirs agentically imagine the possibilities of transformative emancipatory avenues.

*4.3. Emancipation, Reflexivity, and Transformative Communication*

While the Bauls try to challenge and question the mainstream beliefs and practices, their songs seek to raise consciousness among their listeners to create alternate avenues for social transformation. Another important aspect of Bauls is that, they are not just a group of singers; they follow a certain introspective and ideological path, and seek to inspire and teach the common people. In other words, their deep commitments, sacrifices and simple lifestyle, spiritual and discursive practices, attract common people, which eventually gave them credibility at the grassroots level. As a part of their reflexive practice, they constantly question their own identity, privilege and actions. In this introspective engagement, they also question and challenge the existing social systems, norms, and doctrines. According to them, such a reflexive engagement is a continuous and ongoing process, which is also essential for engaging with communities; Sanatan Das Baul, a senior Baul, while performing in an ashram, sang,

> If you are searching for The Human (superior sense),
> Then you worship the humans,
> In everyday practice, offer homage
> To the feet of your human-guru,
> Be a human, with the humans.
> The Human is always creating consciousness,
> The Human resides inside all humans.

Therefore, to become a conscious human being, and to raise consciousness among others is an important aspect of Baul practice. Through constant engagement and interaction with the humans, and through exhaustive introspection, Bauls pursue their journey for knowledge and emancipation.

While calling for love and consciousness, the Bauls also exhibit their commitment in foregrounding broader social inequities and injustices. For instance, while addressing key philosophical debates or spiritual *tattwas* (e.g., *atma-tattwa*, *deha-tattwa*, *guru-tattwa*, *param-tattwa* and others), Baul Shah Abdul Karim remained committed to raise voices for the lower class and lower castes of society. His Baul songs express solidarity with the struggles of the marginalized and outcasts in co-creating a framework for building a peaceful and equitable world.

> The sensitive poets are they who sang the songs of tattwas (theories and epistemologies),
> In my painting, I portray this world's misery and grief,
> And, the demands of the displaced,
> Karim wants a peaceful co-existence, (Majid 2013)

Karim's utterances represent the social and political commitments of Bauls to create a humane world.

Coming from the lower strata, Shah Abdul Karim never forgot his commitments, and never stopped to fearlessly foreground the oppressed and underdeveloped situations of the underserved. Karim further noted that organized efforts from the margins based on solidarity would open up transformative possibilities in underserved spaces. He called for united agentic efforts involving underserved stakeholders to overcome contextual adversities. In one of his songs, he said,

> Oh, my farmer brother, plough the grounds with hands stern,
> With able care and immense passion, treasures we will unearth,
> Saving lives, giving life, is our primary conduct.
>
> The call for produce is upon us, let us brothers join hands,
> Let us wrestle with the earth, to beget Nature's bounty grand.
>
> Farmers and laborers us all,
> Cradled in our Mother Bengal's loving bosom,
> Through hard work and perseverance, cultivate we can gold,
> Alas, but even then, we are stricken by sadness bold.
>
> Farm fish, plant trees, grow much vegetables,
> Reap jute, pluck cotton,
> Grow wheat and grow mustard
>
> Farmers and laborers,
> Fisher-men and handloom weavers,
> Work to your fullest,
> Put all your emotional strength,
> Baul Abdul Karim says,
> Sans this there is no other way.

While most of Karim's songs are about spirituality, he, as a Baul, always remained mindful about his wider societal commitments and the needs of the hour, such as building communities and the nation at large.

In addition, as a part of their spiritual practice, they regularly engage themselves in theoretical discussions and introspections related to their values, actions and philosophies. Thus, through their reflections and performances, Bauls create space to enact their agencies (individual and collective), and search out alternate paths for emancipation. Referring to a famous nineteenth century Baul master, Lalon Fakir, and his humanistic teachings, a contemporary poet, Arun Chakraborty commented in a mediated interview, "In Lalon's philosophy, he speaks of *manobotabaad* (humanity and humaneness). In his approach there is no discrimination based on race, caste or class. That is why his philosophy was humanist one."

Serving and worshiping humans and humanity, Bauls want to create a society, which is just and harmonious (Chakraborty 2001). While narrating about Bauls' spiritual commitment and reflexivity, a Baul from the Burdwan district of West Bengal sang—

> Being a human, respect other humans
> Being a human, know other humans
> Being a human, discover other humans
> Human is the eternal treasure, search for that 'Human' (the man of the Heart).
> To search for the man of the heart is the destination by itself.

Respecting (or worshipping), caring, as well as being human is a continuous and lifelong journey for the Bauls. In other words, teachings of the Bauls fundamentally seek to pose questions as to our perception, preoccupation and inconsistencies towards searching for transformative avenues.

Jha (2010) showed that songs of Bauls are equivalent to hymns and chants, uttered in local dialects and/or mother languages primarily for the purpose of meditations. He further noted, music is just one aspect of broader Baul philosophies; and added that many Bauls do not sing, and only follow the path of their gurus. As Akkas Fakir, a Fakir from Nadia, West Bengal, explained during a mediated interview,

> Music is the medium through which we meditate, our music is very introspective. Our songs are about humanity. You must know to give respect and love to the one who is poor and downtrodden. The songs and teachings essentially question our ego, pride and attachments, which are hindrances to our ways of realizations and journeys to advancement.

Their songs also communicate broader social and ethical commitments towards building a better future. For a Baul, it is important to transcend the boundaries of ego, sense of achievement, social hierarchy, and unearned privileges (like gender and caste). Subhadra Sharma, a woman Baul from West Bengal reflected,

> Boundaries are not only of caste, religion and sect . . . Even knowledge, music, pride, gender . . . all these create limits of identity. If I think I am a great singer, I get trapped in that identity. Some think 'I am an expert or a great devotee', they get shackled by that. The pride of caste, the pride of beauty, the pride of knowledge, the pride of greatness, and the pride of youth (Jouban) . . . these are the five sources of sin. That is why our spiritual path is beyond any caste, religion or class. A Christian, a Muslim, a cobbler, a sweeper (a laborer who sweeps roads and other public places) everyone is treated as a human being in this path. For a state of spontaneous simplicity, all these bounded identities have to dissolve.

Her comments depict that Baul discourses, philosophies and practices are fundamentally grounded in the principles of reflexivity and introspection. It is the engaged realizations and involvements, which characterize their journey towards humane transformations.

In spite of all kinds of mainstream oppositions, Bauls talked about their commitment to follow the path of singing and spreading the message of humanity. Sufia Begam shared her life-story, how as

a women Baul she fought against patriarchal or mainstream religious values, and embraced the path of sacrifice and commitment,

> There was a Fakir residing near our house. He used to regularly perform in the *akhara*. I was drawn to him by his singing. I went and asked him to teach me. He asked me, "Are you sure you want to learn singing? Your father does not sing, you mother does not sing either, nor do your brothers, or anyone in your family." I told him I was serious about learning music. He then told me "Your songs will be invaluable. Yet no one will appreciate you." I told him, "Let it be so. Let no one appreciate me. Yet, I want to learn music from you." . . .

> People came up with a lot of comments. My mother beat me up. My father threw me out of the house. And yet, I was not deterred. I was in love with music. I thought if I could sing my life would have some meaning and purpose to it. I was quite stubborn in my pursuit. And that is how I entered the world of music.

Her words conveying the committed struggles of Bauls that take place at the margins; she also indicated that for sustaining their tradition, Baul teachings are communicated from one generation to the other.

Another aspect of Baul and Fakir practice is not to be fully dependent upon the dominant financial institutions including state (having said that, we are not arguing that they do not need financial support from the state or Non-Governmental Organizations, but historically the rural Bauls tried to survive with alms and support of local people and their followers), but to create a community of admirers to ensure their spiritual journey is sustainable. For instance, following the practice of *Madhukari* (ritualistic begging), they go door-to-door to reach their followers and make regular contacts with followers in local communities; the process is an important one for both their spiritual and physical sustenance, anchoring spiritual bonds in community ties. During *Madhukari*, they perform publicly, and accept any offerings as per the ability and/or choice of their admirers; this practice helps them to reduce their ego. One of the Baul practitioners explained, "One of the daily routines of the Bauls is to go around the village, waking up the villagers by singing. In exchange the villagers used to give the Baul money or grain." Apart from everyday *Madhukari*, Bauls organize spiritual events and gatherings in their *ashrams/akharas* with financial support of followers and well-wishers. One of the organizers of an ashram of Birbhum, India, said, "The (Atal Behari) Ashram runs by the donation of people. We beg door to door for six months to arrange this festival. We serve the mendicant people like sadhu, saints, and others." Thus, local organizing and interactions (as opposed to institutional grants) are also the way for many of them to spread the words of duty, love and harmony. In a documentary—Sama: Muslim Mystic Music, an interviewee commented, "They organize their performances by collecting money themselves. Hence, they do not require (state or external) patronization. The absence of patronization is perhaps why the form has stayed so pure." In other words, their reliance and confidence in local support and patronage help them to keep their discourses, values and ideology out of the reach of the dominant influence or prevent them from getting co-opted (to a large extent) by the mainstream religious and socio-political institutions.

Several scholars and Baul performers argue that Baul and Fakir songs, philosophies and paths offer hope for the coming days, where the world is becoming increasingly insane and turbulent. Choto Golam Fakir, a performer from Murshidabad, India, during an interview commented, "World peace can come through music. No matter what people were made for, the ultimate goal is attaining humanity. Our music has an essence of universal love and peace. If these songs spread across the world—I believe people will live happier lives." Through their discourses, the Bauls talk about global issues from their local spaces, which also opens up scopes for local global exchanges.

In the imaginings of a better (more tolerant and habitable) world, Bauls call for breaking the boundaries of religion, caste, class and thereby seek to build a humane society free from oppression, ostracism and material-scarcities.

> Come be that day,
> When Hindu, Muslim, Christian and Buddhist,

> Divide of caste, class or faith would cease to exist,
> Such a reformed humane society,
> Oh my mind, when would its creation come to be?
>
> When temptations of greed would hold no sway,
> When to take up the shoulder satchel, none will need,
> When no one will shove us apart, Calling us 'an unreformed awful lowly lot!'
> Oppressive shackles will not make us feel continually alienated,
> Such a reformed human society,
> Oh my mind, when would its creation come to be?
>
> The rich and the poor, together, under one common roof would reside,
> Each one will get what each one's deserving shares strike.
>
> Over religion, caste, faith or creed,
> No one would make an upheaval, no one would fight.
> Weeping, says Lalon Fakir,
> Who will be there to show me,
> Such a reformed humane society,
> Oh my mind, when would its creation come to be?

In today's turbulent world and uncertain existence, their songs remind us to serve and care for humans, which is the ultimate aim of the Bauls. It is remarkable that being at the margins of the global South and constantly fighting with contextual odds, never losing faith in transformative capabilities of human agencies, and never stopping to dream about emancipatory possibilities, which we (as a human race) need to ensure together.

## 5. Discussion

Owing to historically-constituted hegemonic oppression and discrimination (both structural and communicative), the discourses (voices and songs) of the Bauls are delegitimized in dominant discursive spaces. Hegemonic punitive policies (e.g., *fatwa*) rooted in the dogmatic behavior of mainstream religious institutions, as well as ethnocentrism in the upper castes and/or classes in society, constitute the material and discursive marginalization of syncretic cultural practices (Das 2014). As a part of the process, the hegemon on one hand diluted and deformed the voices of dissent, and on the other hand, strategically misrepresented the Baul community as heretics and infidel (Mamoon 2008). The marginalization of Baul cultural spaces is further reproduced through the co-optive practices of urban incorporation and commoditization of Baul music.

The impact of dominant discriminations and structural absences are evident in the lives of Bauls and Fakirs. Along with ongoing practices of stigmatization, they are constantly struggling with hunger and poverty. Being a part of an unorganized sector with no social security, the Bauls are fighting an uneven battle with uncertainty, illness, and anxiety. Situations of elderly and women Bauls are particularly precarious, constituted amidst structural transformations of the Indian political economy rooted in market logics. While elderly Bauls are fighting for basic minimum subsistence assistance and health services, the women Bauls are battling for recognition and equality amid patriarchy and religious orthodoxy. Having said that, it is worth noting that a few fortunate Baul performers received attention and financial benefits from the urban patrons and the governments, incorporated into the neoliberal structures of mobility; this paper investigates the condition of common Bauls and Fakirs who live in the lowest socioeconomic strata of rural Bengal. Moreover, this marginalized population of Bauls is experiencing threats from fake Bauls and from market-dictated uneven competition (in the domain of folk music) in that region (Knight 2010). For example, in Bangladesh, resourceful and technology-savvy music bands, chiefly constituted of urban youths, are gaining fame and money by performing Baul songs (sometime by distorting actual tunes) written by Baul Shah Abdul Karim,

when the direct disciples of Baul Shah Abdul Karim namely Ronesh Thakur (and Late Ruhi Thakur) are forgotten.

Scholars such as Openshaw (2002) notes that Bauls are oftentimes portrayed by the hegemon as *deprived of agency*. Harter et al. (2009) argue that for marginalized artists like Bauls, who were excluded from the mainstream socio-political space, aesthetic performances serve as their alternate avenues to reach wider audiences. Bhattacharya (1969) stated that, "socially the Baul tradition can be interpreted as a series of rebellions by isolated individualists against the caste and class system" (p. 30). In many ways, Bauls are the 'stubborn bedrock' (Scott 1990, p. 273), who perform mundane and/or quotidian agentic acts to ensure a long term survival, as opposed to directly winning some 'set-piece battles' (e.g., against oppressions or intolerance) (Scott 1990). This paper argues that it is the ceaseless agentic efforts and abilities of the Bauls and Fakirs, which are the foundations of their consistent struggle against contextual odds and adversities.

For instance, they maintain regular interactions with both broader communities and followers to communicate their messages, as well as travel to the doorstep of the audiences to build a network of solidarity. Such communicative practices are the building blocks of combating their threatened identity, and ensuring the sustainable existence of their spiritual tradition (Krakauer 2016).

Being cognizant about the history and trauma of social exclusion, the Bauls raise their voices through songs to connect to the wider audiences, as well as to organize the community of their followers and admirers. To build community, Bauls practice both intra- and inter-community communication, as well as intergenerational communication. Through such communication, fundamentally, the Bauls question the very structure, make-up, logics and the discourses of dominant society (Sengupta 2015). They essentially ask these uneasy questions with reflexivity and/or by engaging in a continuous introspection process (Liton 2013). On an individual level, they call for questioning our own egos, ethnocentrism and privileges toward developing humane qualities. Again, on a societal level, they diligently work towards raising societal consciousness towards imagining a free and just society. Therefore, the journey is a never-ending process of challenging and improving inner (self) and outer (society) space for a Baul.

From both critical/cultural and postcolonial perspectives, the journeys of Bauls and Fakirs are relevant in this contemporary era for several reasons. Questioning and resisting hegemonic and neoliberal praxis, Bauls call for religious and societal harmony and peace in both local and global spaces. Through discursive engagements, they on the one hand dismantle the hegemonic constructions that portray Bauls as inferior or deviant, and on the other hand, they also question socio-cultural hierarchies such as caste, gender, and class to facilitate material and structural changes. In other words, in their skeptic and non-conforming yet humane journey, their communicative engagements seek to recover delegitimized voices and agencies. As opposed to profiteering, popularity-based and market-centric praxis, rural Bauls are still operating with limited material and communicative resources. For example, they seek to keep their discursive and spiritual traditions alive through regular practice (including studying and debating) and performances. In addition, in terms of finance and resources, they rely on local and community patronage and assistance; many followers and supporters of Bauls believe that such organizing is instrumental in keeping their culture pure and untainted (from hegemonic influences). Through such practice and knowledge production from below, Bauls not only foreground experiential realities at the margins, but also communicate alternate narratives of hope and transformation.

In their anti-caste and pro-human articulations, they consistently exhibit their ideological rebellion and non-conformist commitments (Mahbub-ul-Alam et al. 2014). Spiritual voices of Bauls from the grassroots, and their communication-centric organizing from the marginalized spaces of the global South, open-up alternate avenues to challenging and reshaping socio-political power-dynamics. Through privileging anti-hegemonic voices and anti-orthodox performances, and legitimizing cultural poetics, Bauls legitimize their agency as an anchor to raising consciousness and bringing about transformation. For instance, on one hand, they foreground their experiential narratives of hunger, poverty, mental stress as well as incidents of oppressions to create openings for change; and on the

other hand, they sing songs of emancipation towards transforming structures of the society both materially and discursively. Humane and emancipatory pronouncement of Bauls is therefore both an inner journey of spiritual consciousness, and an outward journey of transforming structures.

**Author Contributions:** Conceptualization, U.D., M.J.D.; writing—original draft preparation, U.D.; writing—review and editing, M.J.D.

**Funding:** This research received no external funding.

**Conflicts of Interest:** The author declares no conflict of interest.

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
