# Peer review of "Songs of the Bauls: Voices from the Margins as Transformative Infrastructures"

_religions, doi:10.3390/rel10050335_

Round 1

Reviewer 1 Report

The article addresses the important topic of a religious music of socioeconomically marginalized people, especially songs addressing poverty and related marginalization. I recommend the well-written and thoroughly thought-through article for publication pending minor revisions.
My suggestions for the minor revisions are as follows:

-Any description of the musical sound is entirely missing. Some description of the sonic should be present. If musical sound is not a focus, which it seems not to be, why that is so should be specified, perhaps in the Methods section. The authors should explain why representations made through musical sound are not taken into account in the study.

-Although online videos are studied, any description of the videos’ contents and visuals is minimal. More of such a description should be provided to give the reader more context.

-A large number of online sources were surveyed for this study. The article would be enhanced by some basic statistical analysis around the following questions. Most importantly: How many songs were found on each of the song lyrics’ topics or themes discussed by the authors? Also: It is mentioned that there are few songs on poverty, but how few in comparison to other themes?

-Can online citations be provided for each of the songs discussed? Right now only academic publications are cited in the bibliography.

Author Response

By paying sincere attention to your valuable comments and feedback, we have modified the manuscript as follows:

(i) Based on your suggestions, we have modified the methods section and mentioned why the manuscript primarily focus on the discourse (or discursive analysis), and not on musical sound (p. 5),

(ii) According to your observations, we have added texts in several places to describe contexts by paying attention to visuals of the videos,

(iii) Attending to your comments, we have added a few statistics on (a) number of songs for each theme (p. 6), and (b) percentage of songs represent poverty and conditions of marginalization (p. 2 and 4), and

(iv) Following your feedback, we have added new citations on mediated resources used in the manuscript.

Reviewer 2 Report

This study examines how bauls—rural minstrels from Bengal who are also viewed as a religious sect by those outside their community—frame and address their social, economic, and spiritual marginalization through song. This study seems to be co-authored by communication studies scholars, who attempt a postcolonial intervention on their subject to examine how the bauls critique frameworks of neoliberal development in their songs. They employ aspects of grounded theory analysis towards this objective.

This subject and approach is laudable; however, as it stands, the article falls short of its promises and needs significant revision before it is ready for publication. One accepted aspect of grounded theory is actually foregrounding the theory itself. That being said, the authors could do much more theorizing for their readers by explaining the significance of the material under analysis. For example, I enjoyed having the translations of baul songs and longer transcripts of exchanges in the documentaries in the article; however, I often felt that the authors assumed the reader would understand the connections they were making rather than being explicit about their connections. For example, section 4.1 does not have a conclusion; it ends with a translation of a text from a baul song (lines 340-344) and the reader is left wondering how this song exemplifies “the endangered existence of bauls” (lines 336 and 337). After a long transcript of a duet, the authors simply tell us to “note here the juxtaposition” (line 379-380) instead of telling us the significance of this exchange. On that note, a more robust article conclusion is necessary: How do bauls raise social consciousness through voice? Why should scholars pay attention to this? How does this connect to the cultural communicative lens or what does this mean for postcolonial scholarship that the authors laid out in the beginning of the article? These questions linger unanswered at the end of the article. Consequently, the article lacks a “so what” quality to it. In other words, the authors need to gain a better control of their argument and analysis; they should be more transparent about how they came to their conclusions.  

Since the authors employ grounded theory and claim to take a critical communicative lens, I would remove “ethnomusicology” from the keywords of the article.Ethnomusicologists always take ethnographic approaches to their subjects, a key component of which is personal interactions with living, breathing interlocutors. While the authors may be from India or Bangladesh (since they mentioned translating baul song materials themselves), they do not place themselves in their narrative and give no indication of interacting themselves with the baul musicians whose material they work with. Another aspect of ethnography is thick description and not letting theoretical exposition overshadow key interlocutors. Because the authors did not use ethnography, we have no thick description of any of the musicians in the article, or the circumstances in which they live—all information comes from the song texts or secondary sources. Another key aspect of ethnography involves scholars placing themselves in their written narratives to maintain reflexivity, with the understanding that they are creating a representation of reality from a specific standpoint. Constructivist grounded theory (what Charmaz favors in the cited 2000 article) also understands that data are reconstructions of experiences, not the experiences themselves. The authors do reference key ethnomusicological works on baul music, but that does not make their own work ethnomusicological. Hence, I believe having the term “ethnomusicology” as a keyword is misrepresentative, and opens the authors to methodological criticisms they are unprepared to deal with.

On a sentence-level note, the authors should remove all uses of the forward slash (/) in their writing. Its use makes the article read too much like a draft, where the authors seemed unsure of which word they want to use. Most of these pairs are synonyms; hence, it would be much less distracting to the reader if one of each pair were removed. Additionally, the authors may want to examine all dyads (uses of “and” and “or” for pairs) as too many bog down the reading experience.

One extremely confusing sentence should be revised or removed: lines 44 and 45 with the Harrison (2013) citation. The ethnomusicologists in question did not sing any songs in this situation—these are scholars, not an ethnic group. I’m wondering if the authors meant the gaine of Nepal, who are referenced in Harrison’s article? Also, I was not sure where the reference to Nigeria came in, because Harrison (2013) makes no mention of Nigeria in her article

Author Response

Based on your valuable suggestions and comments, we have modified the manuscript as noted below:

(i) Attending to your comments, we have modified the methods section,

(ii) Focusing on your observations, we have revised the entire results section; more specifically, we have added texts to elaborate on the contextual details and establish connections. In addition, we have included a brief conclusion for section 4.1 (p. 10),

(iii) Paying close attention to your comments and advice, we have thoroughly modified the conclusion section (p. 19-20),

(iv) As per your suggestions, we have modified the results section to enhance the quality of arguments and analysis,

(v) We fully agree to your valuable comments on ethnomusicology. Accordingly, we have deleted the keyword,

(vi) According to your suggestions, we have edited all the forward slashes (/) and removed the synonymous word-pairs, and

(vii) We are sorry for the erroneous statement; and as per your advice, we have modified the statement (p. 2).